

# Tipping Point Detection and Early-Warnings
# in climate, ecological, and human systems

Vasilis Dakos[1*], Chris A. Boulton[2*], Josh E. Buxton[2], Jesse F. Abrams[2], David I. Armstrong McKay[2,3], Sebastian Bathiany[4], Lana Blaschke[4,5], Niklas Boers[2,4,5], Daniel Dylewsky[6], Carlos López-Martínez[7,8], Isobel Parry[2], Paul Ritchie[2], Bregje van der Bolt[9], Larissa van der Laan[10], Els Weinans[11], Sonia Kéfi[1,12]

[1]ISEM, Univ Montpellier, CNRS, EPHE, IRD, F-34095 Montpellier, France;
[2]Global Systems Institute, University of Exeter, Exeter
[3]Stockholm Resilience Centre, Stockholm University, Stockholm, Sweden
[4]Earth System modeling, School of Engineering and Design, Technical University Munich, Munich, Germany.
[5]Potsdam Institute for Climate Impact Research, Potsdam, Germany.
[6]Department of Applied Mathematics, University of Waterloo, Waterloo, Ontario, Canada
[7]Signal Theory and Communications Dept., Universitat Politècnica de Catalunya UPC, Barcelona, Spain;
[8]Institut d'Estudis Espacials de Catalunya IEEC, Barcelona, Spain
[9]Environmental Sciences Group, Water Systems and Global Change, Wageningen University and Research, Wageningen, The Netherlands)
[10]Niels Bohr Institute, University of Copenhagen, Copenhagen, Denmark)
[11]Copernicus Institute of Sustainable development, Environmental Sciences, Faculty of Geoscience, Utrecht University, Utrecht, The Netherlands
[12]Santa Fe Institute, 1399 Hyde Park Road, Santa Fe, NM 87501, USA

*Corresponding author: Vasilis Dakos (vasilis.dakos@umontpellier.fr), Chris A Boulton (C.A.Boulton@exeter.ac.uk)

**Abstract.** Tipping points characterize the situation when a system experiences abrupt, rapid and sometimes irreversible changes. Given that such changes are in most cases undesirable, numerous approaches have been proposed to identify if a system is close to a tipping point. Such approaches have been termed early-warning signals and represent a set of methods for identifying statistical changes in the underlying behavior of a system across time or space that would be indicative of an approaching tipping point. Although the idea of early-warnings for a class of tipping points is not new, in the last two decades, the topic generated an enormous amount of interest, mainly theoretical. At the same time, the unprecedented amount of data originating from remote sensing systems, field measurements, surveys and simulated data, coupled with innovative models and cutting-edge computing, has made possible the development of a multitude of tools and approaches for detecting tipping points in a variety of scientific fields. Yet, we miss a complete picture of where, how, and which early-warnings have been used so far in real-world case studies. Here we review the literature of the last 20 years to show how the use of these indicators has spread from ecology and climate to many other disciplines. We document what metrics have been used, their success as well as the field, system and tipping point involved. We find that, despite acknowledged limitations and challenges, in the majority of the case-studies we reviewed the performance of most early-warnings was positive in detecting tipping points. Overall, the generality of the approaches employed - the fact that most early-warnings can in theory be observed on any dynamical system - explains the continuous multitude and diversification in their application across scientific domains.



## 1 Introduction

Tipping points characterize the situation when a system shifts between contrasting system states. Such shifts occur when a threshold is crossed and an accelerating transition "moves" the system from its current state to a contrasting one (van Nes et al., 2016). Given that tipping points are associated with abrupt, rapid and sometimes irreversible changes, numerous approaches have been proposed to identify if a system is close to such points. These approaches are often referred to as Early-Warning Signals (EWS) and they represent a powerful generic tool for detecting tipping points in a variety of systems (Scheffer et al., 2009). The general mechanism behind EWS is that, as a dynamical system approaches a tipping point, it becomes slower at recovering from small perturbations, and this critical slowing down of the system (CSD, (Wissel, 1984)) leaves signatures in the temporal and spatial dynamics of the system (Drake et al., 2020). EWS rely on identifying exactly such changes in the underlying behavior of a system across time or space prior to a tipping point.

Early, after their introduction in the literature, it became clear that EWS did not allow to anticipate all types of tipping points in advance (Hastings and Wysham, 2010), and that they are not unique to tipping point responses but also occur when systems are undergoing smoother transitions (Kéfi et al., 2013). These realisations imply that some shifts (typically referred to as abrupt shifts or regime shifts) may require alternative or additional signals (Boettiger et al., 2013; Dakos et al., 2015). Thus, a rich research program has been triggered in the theory behind tipping point detection and the development of tools (Table 1) for quantifying changes in dynamical patterns of system behaviors that could be used as early-warnings preceding tipping points and abrupt shifts in general. Different terms have been used to describe the great variety of metrics proposed in the literature, like 'early-warning systems' (Lenton, 2013b), "observation-based early-warnings' (Boers, 2021), 'leading indicators' (Carpenter et al., 2008), 'resilience indicators' (Dakos et al., 2015), 'generic indicators' (Scheffer et al., 2015), 'dynamical indicators of resilience' (DiOR) (Scheffer et al., 2018), 'indicators of transitions' (Clements and Ozgul, 2018), 'universal early warning signals' (Dylewsky et al., 2023). In the rest of the paper, we will use the term 'early-warnings' to refer to this whole family of indicators.

Whatever the term used, while early-warnings are well grounded in theory, the challenge remains to apply them to real-world systems. A number of review and synthesis papers have summarized



the theoretical aspects of early-warnings and provided partial accounts of their empirical applications (Scheffer et al., 2012a, 2015; Dakos and Kefi, 2022; Nijp et al., 2019; Litzow and Hunsicker, 2016; Alberto et al., 2021; Bestelmeyer et al., 2011; Lenton, 2013b, 2011). Yet, although the utility of early-warnings has led to early-warnings proliferating beyond ecology and climate and have been applied across a variety of scientific domains, we miss a complete picture of where, how, and which early-warnings have been used so far in real-world case studies.

Here, after summarizing the basics of the theory underlying early-warnings and giving an overview of their taxonomy, we review the literature for the use of early-warnings in empirical studies across all scientific fields. We document what metrics have been used, their success as well as the field, system and tipping point involved. We then classify this information in order to provide an overview of the progress, the limitations and opportunities in the empirical application of early-warnings after 15 years of research on the topic.

**Table 1. Available software tools for the estimation of early-warnings with temporal and spatial datasets.**

| Name | Software | Description | Reference |
|---|---|---|---|
| **earlywarnings** | R package | One of the earliest R packages to calculate model and metric based early-warnings | (Dakos et al., 2012)<br>github.com/earlywarningtoolbox |
| **earlywarning** | R package format | Fits a normal form model with and without a saddle-node bifurcation based on a likelihood approach | (Boettiger and Hastings, 2012b)<br>github.com/cboettig/earlywarning |
| **Generic_ews** | Matlab | Matlab translation from the early-warning signals toolbox in R | git.wur.nl/sparcs/generic_ews-for-matlab/-/tree/master |
| **spatialwarnings** | R package | Estimates spatial warning signals based on spatial statistics and spatial pattern formation | (Génin et al., 2018) |
| **ewstools** | Python package | Python translation of the earlywarnings toolbox, with the addition of deep learning classifiers | (Bury, 2023) |



| EWSmethods | R package | early-warning toolbox, inspired by earlywarnings, that omits model based EWS, but that includes multivariate indicators of resilience | (O'Brien et al., n.d.) |
|---|---|---|---|

## 2 The basics of early-warnings

There are three ways that a tipping point may theoretically occur (Lenton, 2013a). A system may undergo a bifurcation when a parameter (or multiple parameters) in the system changes beyond a critical threshold and the stability of the state the system occupies is lost, thus causing the system to shift to an alternative state (bifurcation-tipping or B-tipping). Noise-induced tipping can occur when a system is shifted outside its stable basin of attraction by some form of stochastic forcing

(N-tipping). A third class, known as rate-induced tipping (R-tipping), occurs when a parameter rapidly changes and the system is no longer able to track its stable state (Ashwin et al., 2011).

The majority of the early-warnings discussed below are primarily developed to detect cases where there is a gradual approach towards a bifurcation-tipping event causing a loss of system state

stability. Rate-induced tipping could also show early-warning (Ritchie and Sieber, 2015). Noise-induced tipping is likely to occur unpredictably, and therefore early-warnings are less expected. However, as a system moves towards bifurcation, noise-induced tipping becomes more likely as it becomes easier for the system to leave its current basin of attraction when it is closer to the bifurcation and this increase in the probability of tipping can be identified through particular early-

warnings (Section 2.3). In a realistic scenario with constant stochasticity and conditions gradually changing, tipping is commonly a combination of a movement towards bifurcation and noise pushing the system before the bifurcation actually occurs.

We hereafter present a representative (but not complete) overview of the mostly-used early-

warnings both theoretically and empirically. These signals can be classified in different ways depending, for instance, on the type of mechanism or tipping point (e.g. CSD-based, non-CSD-based), the type of data used (e.g. temporal, spatial, trait, abundance data), the approach employed (e.g. analysing patterns, fitting models, network methods). In Table 2, we suggest a taxonomy of



early-warnings based on the mechanism and the approach used. We then present their basics

without going into the details. A full description as well as methods to estimate them can be found

elsewhere (Dakos et al., 2012; Kéfi et al., 2014; Scheffer et al., 2015; Clements and Ozgul, 2018; Lenton, 2011;

Génin et al., 2018) and in dedicated software packages (Table 1).

| | CSD-based (~ B-tipping) | non-CSD-based (~ B-tipping/ N-tipping) |
|---|---|---|
| **pattern-based** | variance (temporal/spatial)<br>autocorrelation (temporal/spatial)<br>return rate/time (temporal)<br>detrended fluctuation analysis (temporal)<br>spectral reddening (temporal)<br>variance-covariance eigenvalue (temporal)<br>dynamic eigenvalue (temporal)<br>recovery length (spatial)<br>speed of traveling waves (spatial)<br>repair time (spatial)<br>Discrete Fourier transform (spatial) | skewness (temporal/spatial)<br>conditional heteroscedasticity (temporal/spatial)<br>potential analysis (temporal)<br>kurtosis (temporal)<br>quickest detection method (temporal)<br>Fisher information (temporal)<br>mean exit time-Fokker-Planck (temporal)<br>nonlinearity (temporal)<br>trait statistical changes (temporal)<br>Machine-Learning (temporal)<br>Turing patterns (spatial)<br>patch size distributions (spatial)<br>Kolmogorov complexity (spatial)<br>network–properties (spatial/temporal) |
| **process-based** | generalised models (temporal)<br>time-varying AR(p) models (temporal)<br>probabilistic time-varying AR(p) (temporal) | drift-diffusion-jump models (temporal)<br>threshold AR(p) models (temporal)<br>likelihood ratio (temporal) |

**Table 2. A taxonomy of early-warnings depending on whether the warning is based or not on Critical Slowing Down (CSD). CSD-based early-warnings are mostly associated with bifurcation tipping (B-tipping), while non-CSD-based ones both with B-tipping and noise-induced tipping (N-tipping, see also Sect. 2.1). A second dichotomy is based on the approach: whether the warning is a statistical metric based on the dynamical patterns of the system, or whether it is based on a (as simple as possible) process-model. In parenthesis the type of data (temporal and/or spatial) used to estimate the early-warning.**


## 2.1 Early-warnings based on Critical Slowing-Down (CSD-based)

Most-used early-warnings are based on searching for evidence of 'Critical Slowing Down' (CSD)

in the system. Essentially, as the system is forced towards a tipping point, the state it currently

occupies starts to lose its stability, and the restoring feedbacks that 'pull' the system back to that

state after it is perturbed, start to degrade. This causes the system to respond more sluggishly to

these perturbations, and thus slow down (Wissel, 1984). Figure 1 shows this concept visually using





the 'ball in the well' analogy. When the system is more stable, represented by the well with steeper sides, the recovery is faster as the ball (representing the state of the system) returns faster. A system close to tipping, represented by a shallower well, has a slower recovery as the ball takes longer to

return. Eventually, the restoring feedbacks of the system may become so weak that the stability of the current state may be lost, and the system may transition to a new stable state. Mathematically, CSD occurs as the leading eigenvalue of the system approaches 0 from below. However, in reality we do not have the equations that govern the system's dynamics and as such we have to estimate the occurrence of CSD with methods that aim to infer CSD mostly from the patterns of the systems

dynamics or by fitting very simple and generic process-based models (Table 2).

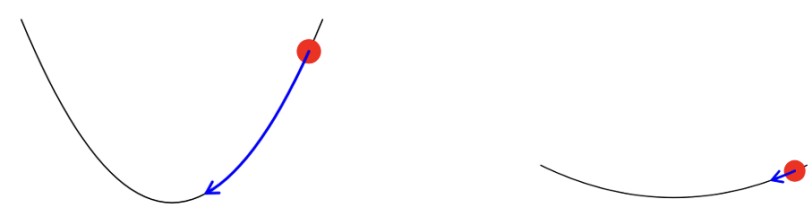

**Figure 1: Using the 'ball in the well' analogy to compare a system that is (left) far from tipping, and (right) close to tipping. The system that is further away from tipping recovers faster from perturbations, the steeper sides of the well describing the stronger restoring feedbacks of the system. Closer to tipping, the sides of the well are shallower, such that the system will take longer to return from the same perturbation because the restoring feedbacks are weaker.**

**2.1.1 Return rate, autocorrelation and variance**

Using statistical techniques makes it possible to detect CSD based on the dynamical patterns a system is generating. The most direct way to detect CSD is to consider the rate at which a system returns to its initial state following a perturbation (return rate or return time). A resilient system with strong restoring feedbacks will return to its initial state faster than one which is near to a

tipping point (Wissel, 1984). However, this method requires the occurrence of a well-defined





perturbation, as well as clear knowledge of when the equilibrium state of the system has been reached, neither of which are always clearly defined in the real world.

As the system approaches a tipping point and its recovery slows down, each time step X(t) is more

correlated to the previous timestep X(t-1) (as shown in Fig. 2). This can be measured with lag-1 autocorrelation, or AR(1), which tends towards 1 as a system experiences CSD prior to tipping to an alternate state. Visually this can be viewed by observing a scatterplot of a timeseries of the system against the timeseries lagged one time point (Fig. 2). When the system is far from tipping (top row Fig. 2), there is no relationship between the system now and itself at the previous time

step (low AR(1)). As the system approaches the tipping point, CSD means that there is a strong correlation between the system now and itself at the previous time step (and thus a higher AR(1)). Larger deviations in the red section of the timeseries can be seen, further showing this slowing down and increase in AR(1).



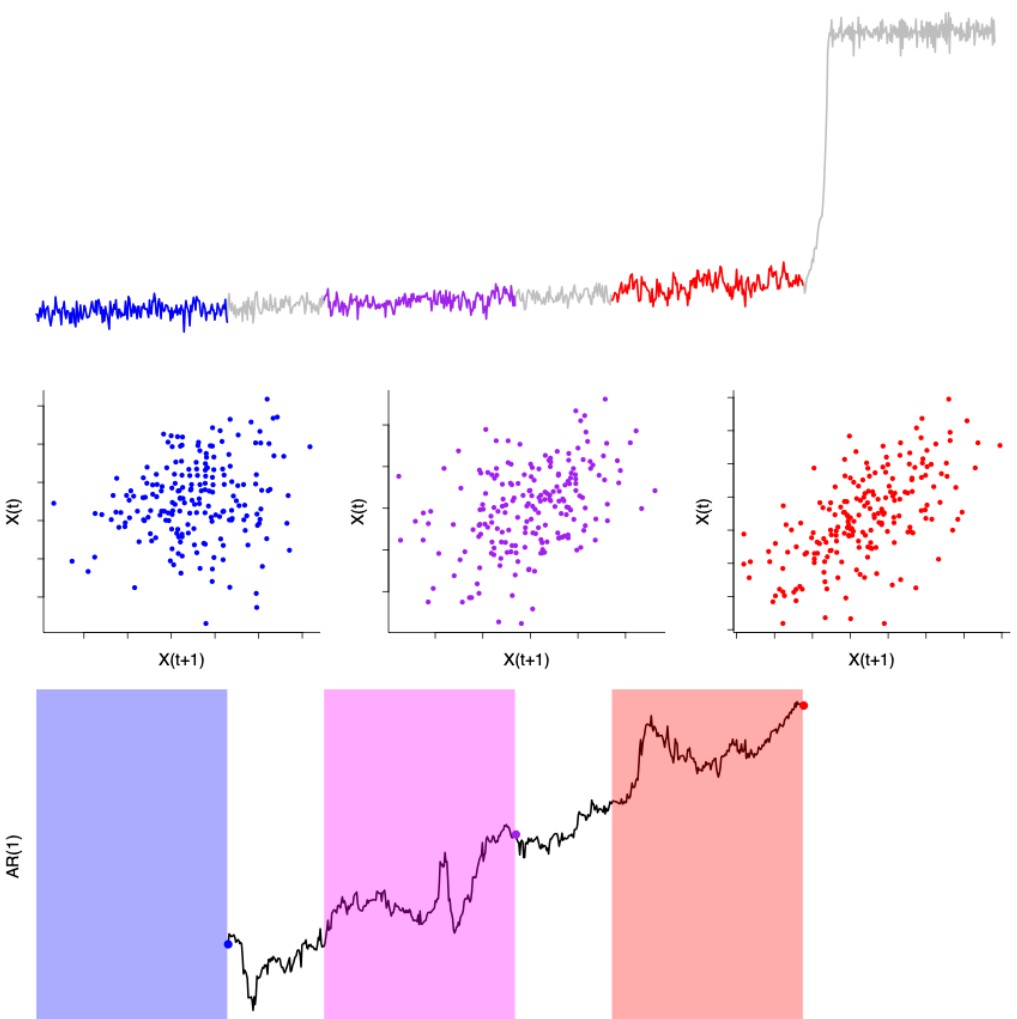

**Figure 2: A comparison of the lag-1 autocorrelation (AR(1)) for a system that is far from tipping (blue), getting close to tipping (purple), and close to tipping (red). As the timeseries approaches tipping (top row), there is no correlation between the timeseries and itself at the previous time point in the blue part of the timeseries, far from tipping. However, closer to tipping, in the purple and then red regions of the timeseries, there are correlations and thus higher AR(1) values. In the timeseries itself there are clear deviations towards the end compared to the beginning, suggesting CSD is occurring as the tipping point approaches. The early-warning are calculated on a moving window (coloured regions in bottom plot). Here, AR(1) is shown at the end of the window used to calculate it, with examples shown as coloured points to match those windows on the detrended timeseries.**

Similarly, as the system struggles to return to its initial state as resilience is lost, the variance of the system is also expected to increase, as the system can sample more of the 'state space' (all the possible states the system can be in) due to the shallower well. However, this is often recorded



alongside an increase in AR(1), because other factors can lead to a change in variance, such as how the system is forced externally.

Spatial analogues of the temporal variance and temporal AR(1) exist too with a similar underlying theory to the one for the temporal ones: as a system approaches a tipping point, and responds more sluggishly to external perturbations and samples more of the state space, it is expected that there will be a higher spatial autocorrelation (Dakos et al., 2010) and spatial variance (Guttal and Jayaprakash, 2009).

Just like AR(1) and variance, all other CSD-based early-warnings aim at detecting characteristic changes in the dynamical patterns of the system either by estimating directly a statistical property (e.g. spectral reddening) or by fitting a statistical model (e.g. detrended fluctuation analysis) (Table 2). A parallel approach involves more complex methods to predict the movement towards tipping points that involve the use of simple process-based models. One such example is that of using a

generalised model that integrates knowledge about the system into a model, which may allow us to estimate changes in the leading eigenvalue of the system, once minimal model assumptions have been made (Lade and Gross, 2012).

**2.2 Early-warnings not based on Critical Slowing-Down (non-CSD-based)**

CSD-based early-warnings rely on the assumption that system dynamics represent deviation around the equilibrium state of the system. However, this assumption doesn't hold in the presence of strong stochasticity. In other cases, either CSD is hard to measure or more idiosyncratic metrics have been suggested to act as alternatives to CSD-based warnings. Below, we outline a few of the most representative non-CSD-based early-warnings (Table 2).


**2.2.1 Skewness**

As the current equilibrium state of the system is losing resilience and the probability to shift to an alternative equilibrium increases, the temporal distribution of states of the system is expected to become increasingly skewed toward the alternative state. This can be quantified by the skewness

of the system. The skewness may increase or decrease, depending on whether the alternative



equilibrium is larger or smaller than the current equilibrium (Guttal and Jayaprakash, 2008). Similarly to the change in skewness observed between the two states with temporal data, it is also possible to observe this change in skewness in the spatial domain (Guttal and Jayaprakash, 2009).

**2.2.2 Flickering**

'Flickering' is the situation where strong stochasticity can "push" a system temporarily into the basin of attraction of the alternative state before returning to the current state with increasing likelihood as the system is approaching tipping (Wang et al., 2012; Dakos et al., 2013). Flickering can be measured either by a simple increase in variance (Dakos et al., 2013), or more complex statistical

approaches (e.g. quickest detection method (Carpenter et al., 2014), heteroscedasticity (Seekell et al., 2012; Seekell and Dakos, 2015)).

**2.2.3 Potential analysis**

Information about a system at multiple sampling points through time or multiple locations across

space can allow reconstructing a 'stability landscape' of the system - or potential, which gives an idea of the more frequent states of the systems observed in systems experiencing different environmental conditions and history (Livina et al., 2010). A multimodality in such a landscape for a given set of environmental conditions suggests that the system could exhibit alternative stable states for that range of conditions (Hirota et al., 2011; Staver et al., 2011; Abis and Brovkin, 2019; Scheffer et

al., 2012b).

**2.2.4 Spatial patterns**

A number of ecosystems have a clear spatial structure, which is self-organised (e.g. drylands, peatlands, salt marshes, mussel beds; (Rietkerk et al., 2008)). Theoretical models have shown that the

size and shape of the spatial patterns change in a consistent way along stress gradients, and as such they are good indicators of ecosystem degradation (von Hardenberg et al., 2001; Rietkerk et al., 2004; Kéfi et al., 2007). Probably one of the most studied examples is the case of dryland ecosystems, where changes in the shape of regular patterns (Rietkerk et al., 2004) and in the patch size distribution (Kéfi et al., 2007) could inform about the stress experienced by the ecosystem (Dakos et al., 2011).




### 2.2.5 Fitting a threshold model

An alternative approach to pattern-based early-warnings is based on fitting process-based models on the timeseries of a system prior to a tipping point. This approach mainly consists of fitting the simplest dynamical model with a tipping point (i.e. a saddle node normal form) <sub>(Ditlevsen and</sub>

<sub>Ditlevsen, 2023)</sub> and testing its likelihood compared to a model without a tipping point <sub>(Boettiger and</sub> <sub>Hastings, 2012b)</sub>. Or fitting threshold models assuming simple autoregressive state-space models <sub>(Ives and Dakos, 2012; Laitinen et al., 2021)</sub>.

### 2.2.6 Structural changes

A novel idiosyncratic way to detect tipping points involves monitoring structural changes properties (e.g. connectivity, node centrality) in network systems (i.e. a network of interacting components) like spatially-connected sites, interacting actors or species in a community <sub>(Mayfield</sub> <sub>et al., 2020; Cavaliere et al., 2016; Yin et al., 2016)</sub>. Alternatively, temporal correlation between components in multivariate systems has been used to construct an interaction network and analyse

its structural properties <sub>(Tirabassi et al., 2014)</sub>.

### 2.2.7 Trait changes

Another idiosyncratic approach involves monitoring changes in the statistical moments of fitness-related traits (e.g. body size) <sub>(Clements and Ozgul, 2018)</sub>. Such trait changes have been found in

populations under stress where changes in the traits of individuals (i.e. decreasing mean and increasing variance in body size) <sub>(Spanbauer et al., 2016; Clements and Ozgul, 2016)</sub>. These trait-based as well as the above-mentioned structural-based signals are case-specific and idiosyncratic to the details of the system as there is no universal mechanism that would generate an expected pattern related to the approach of tipping points.

**3 Overview of early-warnings empirical research in the last 20 years**

We performed a (not systematic) literature review on the empirical (not theoretical) use of warning signals. We first did a topic search (TS) that includes title, abstract and keywords in the Web of Science for the period from 01.01.2004 to 01.04.2023 with the following terms TS=(("tipping



point*" OR "tipping" OR "catastrophic bifurcation*" OR "catastrophic shift*" OR "regime shift*"

OR "abrupt shift*" OR "critical transition*") AND ("early-warning*" OR "early-warning*" OR "warning sign*" OR "resilience indicator*" OR "leading indicator*" OR "precursor*")). We selected as the starting date of our search the year of 2004, despite the fact that CSD was much earlier known in ecology (Wissel, 1984) and signatures of catastrophic bifurcations were theoretically described for dynamical systems (Gilmore, 1981). Our choice was driven by the fact

that 2004 is around the year of the first studies in climate (Kleinen et al., 2003; Held and Kleinen, 2004) and ecology (Carpenter and Brock, 2006) where the theoretical idea of using CSD as warning signals emerged, while few years later the first review on early-warnings on critical transitions was published (Scheffer et al., 2009). Within this time period, our topic search returned 887 unique publications. For completeness, we also ran the same topic search before 2004 going back to 1960,

and we retrieved 11 publications of which only 1 was related to bifurcations. Clearly, we might have missed relevant records with the TS we selected. For example, had we also included the term "phase transition*", we would have retrieved 3,916 records. We decided not to include this term as it pertains to a specific and rich field of physics, but with our TS we are confident to have a rather complete overview of the tipping point (and related terms) literature.


We screened all 887 publications to select only the ones where there was an empirical application of early-warnings (i.e. an indicator was measured on real data to signal the occurrence of a tipping point). This screening led to 229 papers that we classified as ones that have included at least one empirical application of early-warnings. For each paper, we collected the following information:

"domain" (e.g. climate, ecology), "system" (e.g. Arctic Sea Ice, fisheries, mental depression), the "tipping point" described, data source (e.g. lab experiment, field survey, remote-sensed datasets, social data), "data type" (i.e. temporal, spatial, spatio-temporal), "indicator" (i.e. the specific warning signal(s) used), "performance" (whether the performance of the early-warning was reported in the paper as positive, negative, mixed (in the case of multiple signals used or multiple

datasets analysed), or inconclusive). To facilitate the analysis, we regrouped "data-source" and "indicator" categories in broader groups (see appendix X). We also created two extra categories: we classified systems under a specific "field", and we introduced an "indicator type" based on whether the early-warning was CSD-based or non-CSD based. We then excluded the running year



2023 and summarised results in terms of unique publications using simple statistics and alluvial plots in R (4.3.1).

### 3.1 Overall use of early-warnings across disciplines

We were able to classify the total 229 papers published from 2004 to 2023 in 5 main domains: ecology, climate, health, social and physical sciences. We found that empirical papers first

appeared in 2007 in the domains of ecology and climate but only after 2010 and 2011 were the first papers in health, social and physical sciences published (Fig. 3). This change may be associated with the highly cited review by Scheffer et al in 2009 (Scheffer et al., 2009) that introduced (and popularised) the term early-warning signals and critical transitions. Since then the number of empirical studies has quickly increased, but remained dominated by ecology (43.6% of

the papers overall), followed by health (22.6%), climate (14.6%), social (12%) and physical sciences (7.6%) - showing the diversification of the uses of early-warning (Fig. 3).

The higher number of publications in the health domain compared to the climate domain is unexpected. We found a big number of studies in the medical field (Fig. S1b) that form a distinct

group on the emergence of human diseases, such as cancer (Liu et al., 2020), which uses a Non-CSD but rather a context-specific early-warning ("dynamic_network_biomarkers", see also Sect. 3.3). Zooming within each domain, we observed that the most ecological studies are on terrestrial and freshwater fields (Fig. 1Ba) namely on drylands and forests and lake ecosystems (Table S1). The majority of climate studies are on past climate transitions and modern records (Fig. S1c, Table S2),

while the social studies are split between societal shifts (like in politics, social behavior, transport) and finance transitions (Fig. S1d, Table S4). Lastly, studies on physical sciences appear more heterogeneous including tipping points in materials, power systems, or even astronomy (Fig. S1e, Table S5).



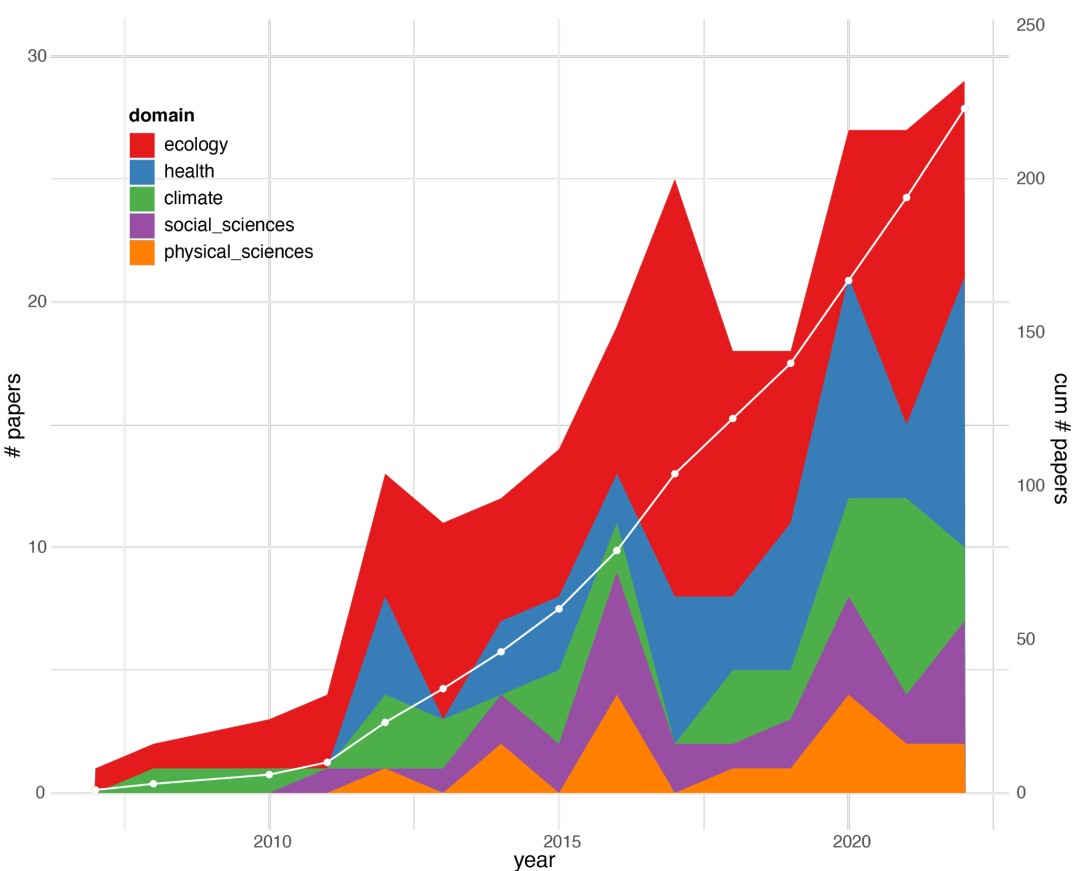


**Figure 3: Evolution of studies applying early-warnings in empirical datasets. The total 229 papers we identified through our literature review between 2004 and 2022 were classified within five main scientific domains (ecology, health, climate, social sciences, physical sciences). White dotted line shows the cumulative number of papers.**

**3.2 Multiple sources of data used**

Across scientific domains, the vast majority of early-warnings were analysed on temporal data (77.7%), while the spatial data were used in only 8% of all studies (Fig. 4) only pertaining to ecology (Fig. S2). Survey data made the majority of the data sources (43.8%; give examples of such data?), followed by data from lab experiments (20.7%), remote sensing (12%), paleo-

reconstructions (10%), and field experiments (7%). This partitioning can be mostly explained by our classification, meaning that we have grouped together a heterogeneity of data sources (e.g. field surveys, historical climate data, social study surveys, hospitalisation records, Supplement A). However, it also reflects the availability of each data source (e.g. most survey and paleo data were



readily available and reanalysed in the context of tipping points), or the difficulty in their

acquisition (e.g. field experiments are harder to execute compared to lab experiments). Looking at
how data sources are used across domains, ecology is the only domain where all kinds of data
sources have been used. What is also interesting to note is that two sources of data are increasingly
used: survey data and remote-sensed (Fig. S3). Specifically, the latter were the latest to be used
(2011), but show a consistent rising pattern over the last years mainly due to the fact that satellite

products span by now a long enough time period (~20 years) to allow the estimation of early-
warnings.

A closer look at studies using remote-sensed products reveals a focus on the analysis of temporal
early-warning on land environments, mainly forests and drylands , but also extended to the

cryosphere, focusing on the analysis of the Arctic and the Antarctic ice sheets (Carstensen and
Weydmann, 2012; AlMomani and Bollt, 2021). The spatial resolution of remote-sensed data has been also
exploited for the identification of spatial early-warning, especially regarding desertification
(Berdugo et al., 2017) and vegetation analyses (Majumder et al., 2019). Overall, we found that the use of
remote sensing products offers two distinct yet complementary approaches to detect early-

warning; high-level products, which correspond to physical variables, for instance sea surface
temperature (SST) (Wu et al., 2015) or different types of indices like the normalized difference
vegetation index (NDVI) (Liu et al., 2019) and low-level products, or direct sensor observables.

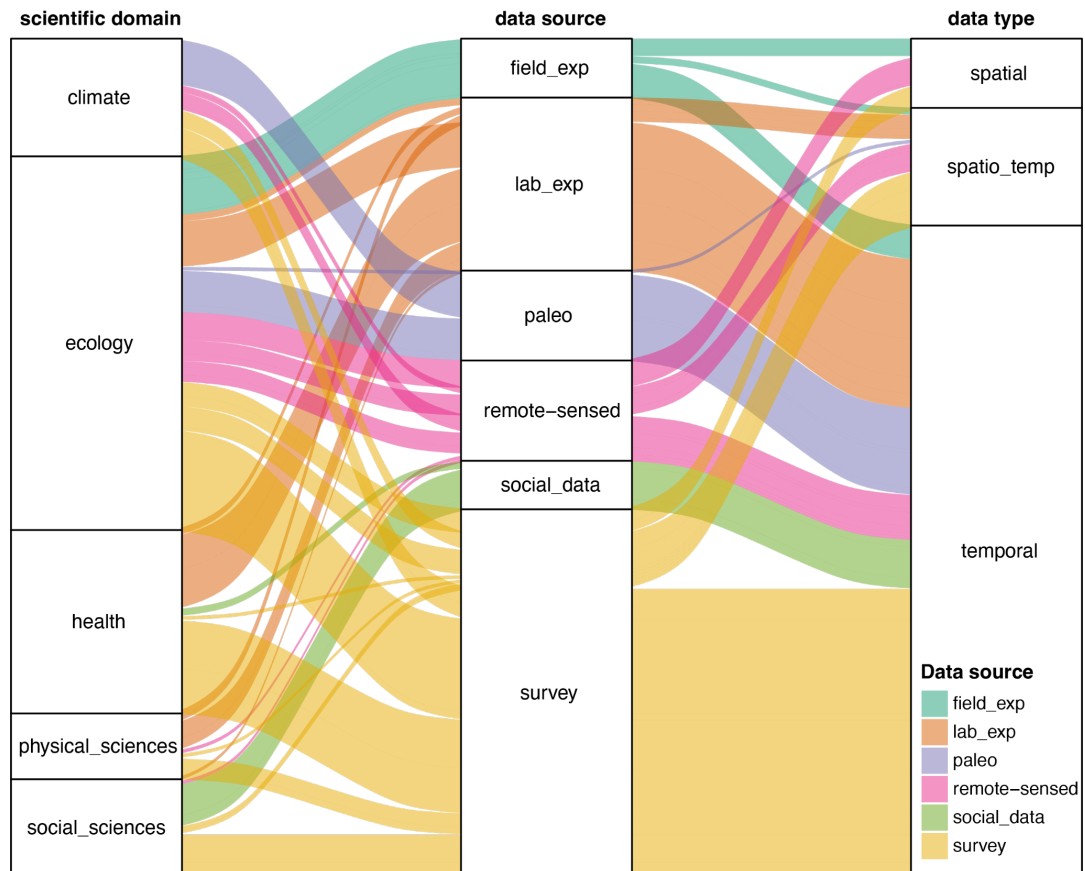

**Figure 4: Alluvial plot connecting scientific domains, data sources and data types. Colors indicate the data source used for the estimation of early-warnings. The size of the boxes in each column represents the proportion of each category. The figure is "read" from the middle column ('data source') to either the right ('data type') or the left ('scientific domain'). The thickness of the lines are proportional to the studies of a given data source that belong to a certain domain (from the 'data source' column to the 'domain' column) or are proportional to specific data type used (from the 'data source' column to the 'data type' column). For example, for the 'data source' field experiment (light green), all studies using field experiments belong to the 'ecology' domain, while field experiments are split into 3 types of data (spatial, spatio-temporal and temporal). [field_exp: field experiment, lab_exp: lab experiment, paleo: paleo-reconstructed data, remote-sensed: data through remote-sensing, social_data: financial data and from social media, survey: data from surveys (field, lab, social)]**

### 3.3 A growing list of of early-warnings

We recorded 65 different early-warnings after reclassifying ones into the same group (for example variance, coefficient of variation and standard deviation were reclassified as "variance", Supplement A). As expected, the majority were CSD-based warnings (74.9%), while 25.1% were non-CSD based ones. Out of the 65 reclassified early-warnings only 21 were used more than once (the rest 44 early-warnings were used only once, Fig. 5, Supplement A). Variance and




autocorrelation were the dominantly used early-warnings across all domains, followed by skewness (Fig. 5). Besides these three early-warnings, the remaining 18 were used selectively within particular domains. The most striking are "spatial variance" (only used in ecological studies) and "dynamic network biomarkers" (only used in health studies, see also Sect. 3.1). Within domains (Fig. S4), ecology is the domain with the highest heterogeneity in the early-warnings (18 out of the 21 used more than once), followed by health (10), and climate (7).

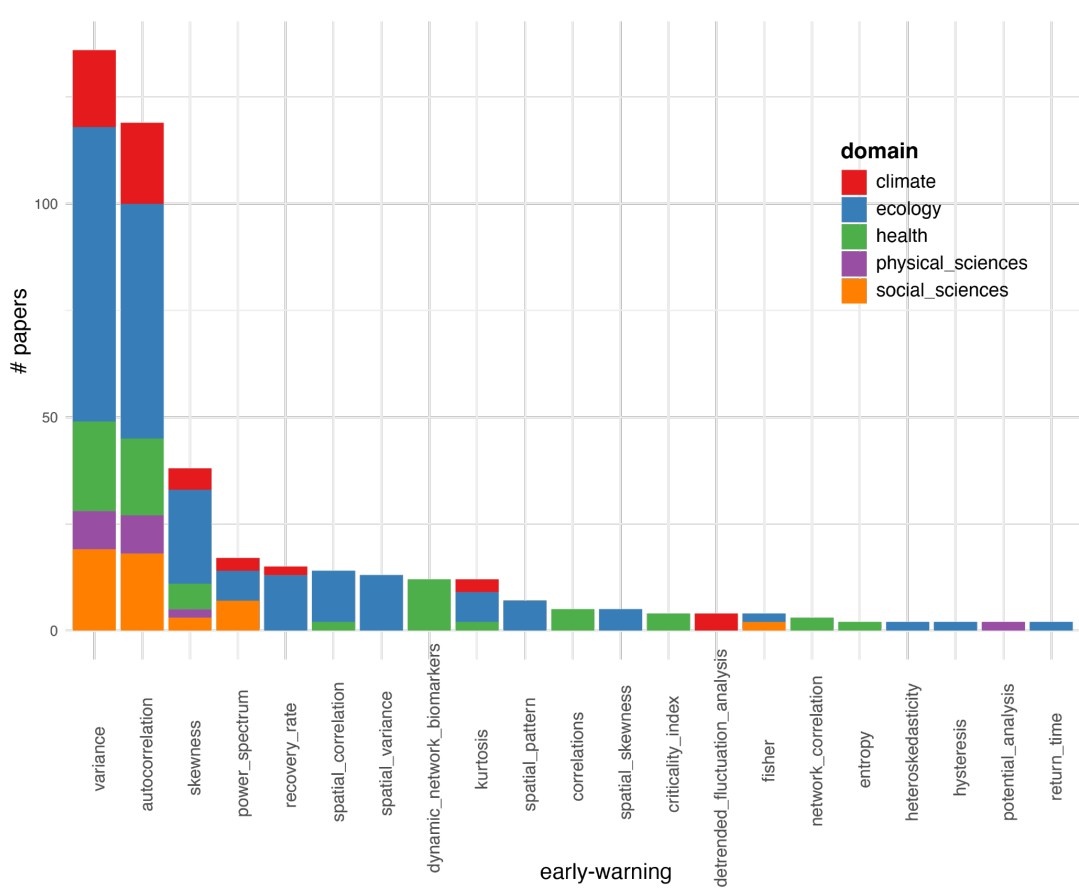

**Figure 5: Number of papers of the 21 early-warnings used more than once in our literature review. Each bar is partitioned into the 5 scientific domains.**



**3.4 A positively skewed performance of early-warnings**

The performance of all 65 reclassified early-warnings (Sect. 3.3) across all domains was positive in 67.8% of all studies (Fig. 6). Only 3.4% of the studies reported negative performances (i.e. no or opposite to expected warning). Ecology reported the most negative results, followed by climate and health, with none reported for physical or social sciences. The performance in the rest of the studies were either mixed (i.e. positive or negative in studies which analysed multiple early-warnings or datasets, 24.7%) or inconclusive (i.e. a statistically weak result, 4.8%). This is an impressively positively skewed result potentially reflecting the known bias in publishing significant results (Fanelli, 2012) or in post hoc analysis where a tipping point has been already documented and early-warnings have been applied in hindsight (Boettiger and Hastings, 2012a).

Interestingly, all the negative results included CSD-based warnings, while for non-CSD warnings only a small fraction reported inconclusive or mixed results (Fig. 6). This difference could be attributed to the fact that non-CSD warnings are at times idiosyncratic developed for the specific system under study compared to the more generic CSD-based. Indeed, focusing on the 21 early-warnings that were used more than once (Sect. 3.3), such system-specific indicators (such as the "dynamical_network_biomarkers") always had a positive result (Fig. 7). Overall, the least-used warnings were associated with a positive performance, whereas the most-used ones (like variance, autocorrelation, skewness, power spectrum) showed all types of performance. There was no early-warning that had predominantly negative or mixed results except for kurtosis (Fig. 7). There was no particular difference in the performance of the more-than-once-used early-warnings across domains (Fig. S5).




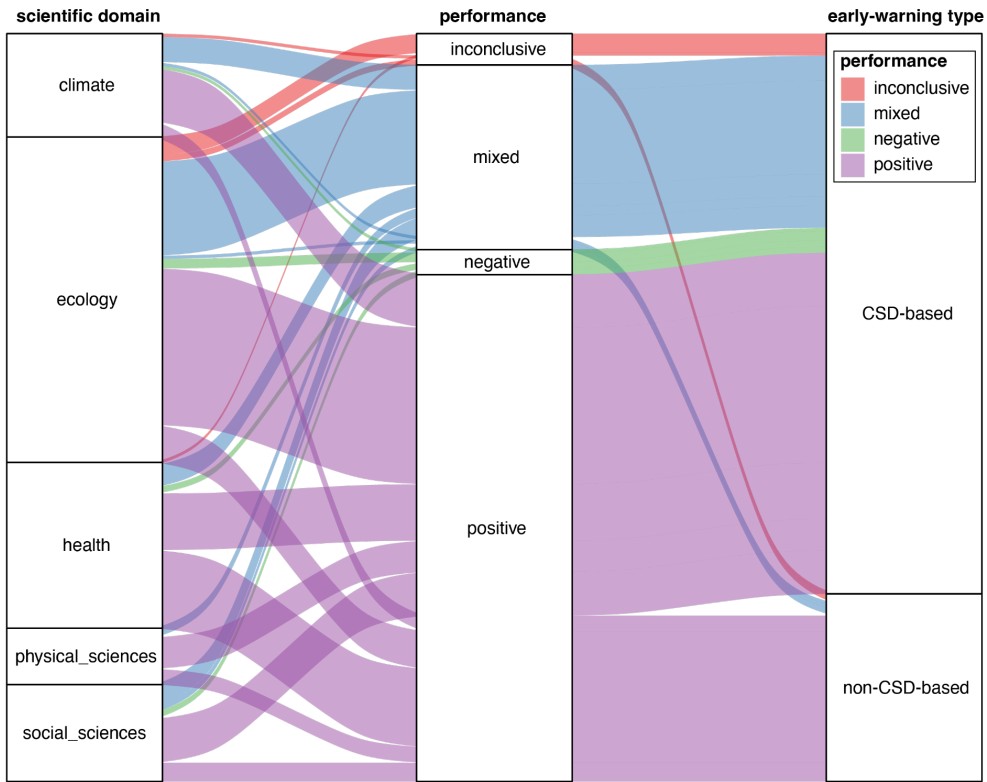

**Figure 6 Alluvial plot connecting scientific domains, the performance of the early-warnings and the type of early-warning (CSD-based vs non-CSD-based). Colors indicate the performance. The size of the boxes in each column represents the proportion of each category. The figure is "read" from the middle column ('performance') to either the right ('early-warning type') or the left ('scientific domain'). The thickness of the lines are proportional to the performance that belongs to a certain domain (from the 'performance' column to the 'domain' column) or are proportional to the type of early-warning (from the 'performance' column to the 'early-warning type' column). For example, for the 'performance' mixed (blue), studies with mixed performance where done with both CSD-based and non-CSD-based warnings ('early-warning type' column), while the CSD-based mixed were found in all domains and the non-CSD-based were split among climate, ecology and social sciences ('scientific domain' column). ['Positive' performance indicates there was a warning identified; 'negative' no warning was identified; 'mixed' indicates positive and negative performances when tested in multiple datasets or when testing more than one early-warning in the same dataset; 'inconclusive' the results could not indicate neither a positive nor a negative warning.]**




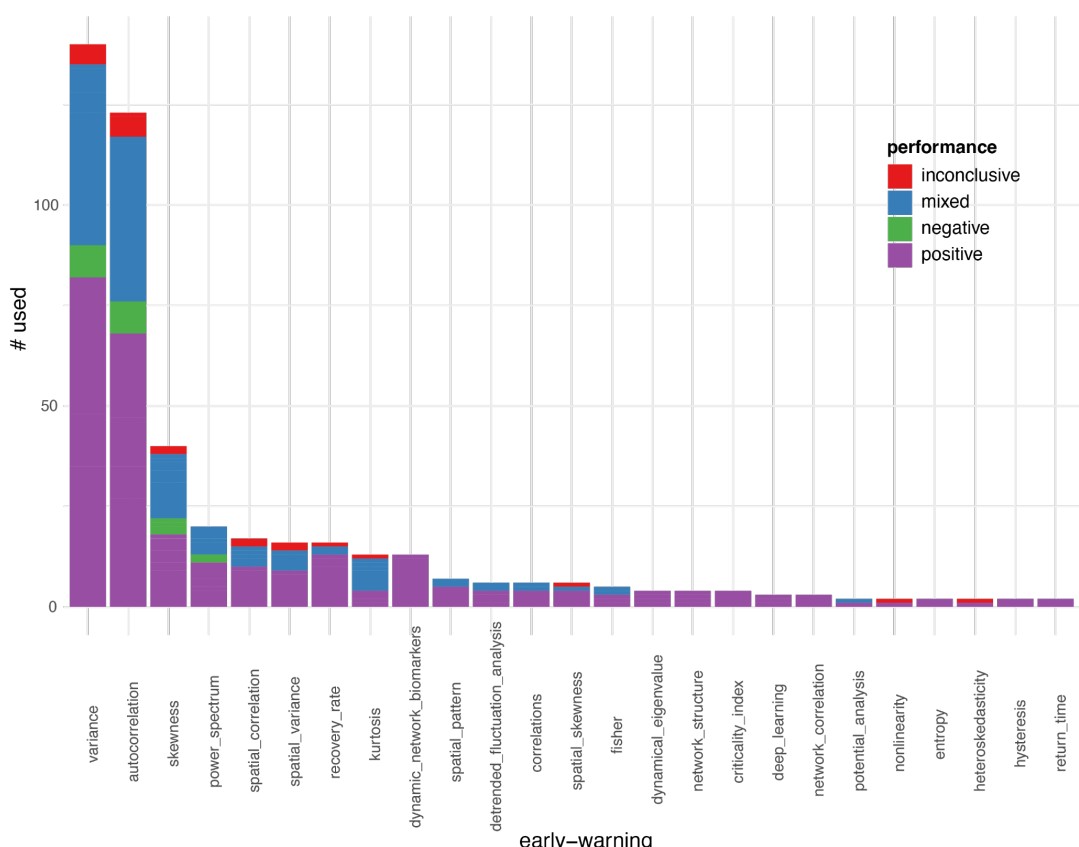

**Figure 7 Early-warnings (used > 1) and their performance. 'Positive' performance indicates there was a warning identified; 'negative' no warning was identified; 'mixed' indicates positive and negative performances when tested in multiple datasets or when testing more than one early-warning in the same dataset; 'inconclusive' the results could not indicate neither a positive nor a negative warning.**

## 4. Discussion

The idea of early-warnings based on CSD is relatively old. In a textbook on "Catastrophe Theory for Scientists and Engineers" from 1981, Glimore already talked about "catastrophe flags" for indicators of CSD (Gilmore, 1981). After an early ecological paper (Wissel, 1984), 20 years later the topic generated an enormous amount of interest, mainly theoretical, with the first empirical tests being on past climate tipping points (Livina and Lenton, 2007; Dakos et al., 2008). Our review of the literature of the last two decades shows how the use of these indicators has since spread to many other disciplines. Indeed, the generality of the approach - the fact that CSD can be observed on any dynamical system, independent of the details of the underlying dynamical equations driving



the system - created an opportunity for testing their validity on many systems and explains the enthusiasm they generated and the diversification of applications which followed.


### 4.1 Early-warning applications: a success story?

Our literature overview suggests that the 65 early-warning signals identified successfully detected a tipping point in almost 70% of the times they were used. This is an impressively positive result, which should nonetheless be treated with caution. First, in many cases, the empirical studies were

conducted on systems that are either relatively simple or under controlled lab conditions, making them mostly proof-of-principle demonstrations (Dai et al., 2012; Veraart et al., 2012). Second, most empirical studies were restricted to hindsight application, meaning that an a priori knowledge of a tipping point may introduce bias towards detecting CSD indicators (Boettiger and Hastings, 2012a; Spears et al., 2017). Third, the documented publication bias against negative or insignificant results

(Fanelli, 2012; Franco et al., 2014) probably applies in the case of the early-warning research given the attention the specific topic has attracted in recent years.

These considerations do not reduce the value and prospect of early-warning research. For instance, one of their biggest values lies in the possible detection of an approaching tipping point. A number

of studies has demonstrated the potential proximity of tipping points in modern climate data using early-warnings (Boers, 2021; Boulton and Lenton, 2015; Ditlevsen and Ditlevsen, 2023). It has also become increasingly clear that early-warnings can be very useful for comparing the resilience of similar ecosystems across space (e.g. (Verbesselt et al., 2016; Forzieri et al., 2022; Lenton et al., 2022) to provide an approximate estimate of resilience that can help prioritise management. In this way, with

'resilience maps', rather than speculating about the proximity to a potential threshold, we can rank situations at a given moment and place.

### 4.2 Challenges

There are a number of conceptual, operational and methodological challenges that can blur the

detection of early-warnings on real data (Dakos et al., 2015). For these reasons, some studies have highlighted their failure at detecting early-warnings on data (Burthe et al., 2016), while others have raised caution about their uninformed use without knowing more about the system's drivers and





underlying mechanisms (Boettiger et al., 2013). In what follows, we discuss some of the most important challenges related to the use of early-warnings.


### 4.2.1 Fast changes, slow responses, stochasticity, limited data challenge early-warning performance

The detection of early-warnings relies on the assumption that the system is approaching a transition gradually. A system should be externally forced on a slow timescale towards the tipping point,

while experiencing perturbations on a shorter timescale such that CSD-based early-warnings signals can be estimated. In theory, it is generally assumed that the short-term noise is independent and identically distributed with a mean of zero. This is unlikely to be the case, with climate systems experiencing extreme weather events, for example, which are likely becoming more prevalent with the changing climate. There have been 3 '1-in-100 year' droughts in the Amazon rainforest since

2005 (Erfanian et al., 2017; Lewis et al., 2011) which clearly alter the signals observed. For cases like these, it is worth measuring early-warnings on the drivers themselves, such as timeseries of rainfall for vegetation systems. If these show early-warnings, then it is likely that signals observed in the system itself are being driven by changes in forcing rather than a movement towards tipping.

When monitoring a system, longer timeseries are desirable to detect the upcoming tipping point. For instance, the best-case studies found in this literature review from remote-sensed products, which have been available since ~1972, have approximately 50 years long timeseries. However, due to sensor degradation and upgrades, it can be challenging to get a long timeseries from a single sensor, and products are often created from combined data sources. This can interfere with most

of the early-warnings, if this merging changes the signal-to-noise ratio (SNR) across time (Smith et al., 2023). For example, newer sensors will measure with a greater radiometric accuracy, increasing the SNR and in turn 'erroneously' increasing the AR(1) as far as an early-warning is concerned. This increase in SNR will also decrease variance, thus allowing the user to check for anticorrelation between AR(1) and variance to see if the early-warnings are being influenced or not.


As well as questions around data availability and noise behaviour, the inherent timescale of the system being studied can hinder our ability to detect tipping points. While tipping is by definition



a 'fast' process, for slow moving systems like the thermohaline circulation (AMOC), this tipping
event occurs over decades and could therefore be difficult to detect that the tipping point has been
passed using early-warnings. Another example of this is the Amazon rainforest, where at least in
modelled vegetation, there is a slow response of the forest based on the climate change that it has
been subjected to (Jones et al., 2009). It could take decades for dieback to occur even under a constant
climate such that a tipping point could be passed long before it is actually realised (Hughes et al.,
2013). This 'committed response' has been explored in a number of GCM experiments (Jones et al.,
2009; Boulton et al., 2017), but it is unclear how early-warnings would be affected by this (Van Der Bolt
et al., 2021).

### 4.2.2 Non-specificity of early-warnings

The generic and universal character of most - but in particular CSD-based - early-warnings comes
at a price of these warnings not being specific to abrupt and irreversible tipping points. Instead
they can be used to also detect smooth and reversible transitions (Kéfi et al., 2013). This limitation
suggests that we need other, additional indicators, in particular system-specific indicators (Boettiger
et al., 2013). In the case of spatially structured ecosystems such as drylands, for example, studies
have shown that temporal early-warnings could fail (Dakos et al., 2011), in which case, the use of
the changes in the patterns themselves could provide a good alternative (Rietkerk et al., 2004; Kéfi
et al., 2007). In the same way, specific indicators have been developed in health sciences for the
monitoring of disease emergence (Table S3).

System-specific early-warnings may also be a better prospect, where understanding about
processes in the system can help us to monitor its resilience in novel ways (Boulton et al., 2013).
Yet, the original idea behind the development of early-warnings was based on the premise that this
knowledge is missing or insufficient and thus a pattern-based approach could be more informative
(Scheffer et al., 2009). Thus, the challenge is to strike the right level of system-specific warnings
and to combine them with the generic ones. For instance, trait-based (Clements and Ozgul, 2016)
and function-based (Hu et al., 2022) warnings have been recently suggested as complementary to
the existing generic warning signals. A first step towards that direction could be to map the 65
classified early-warnings we reviewed on a gradient of generic to system-specific indicators.



### 4.2.3 Multivariate (high-dimensional) systems

Most early-warnings are well-tailored for uni-dimensional systems, meaning systems described by a single observable (e.g. vegetation cover). However, real dynamical systems are typically high-dimensional and the quantification of early-warnings in those multivariate systems presents challenges. For instance, two different variables may give conflicting information, or obscure a clear signal (Boerlijst et al., 2013; Weinans et al., 2021). Thus, it is challenging to know what and

how to measure early-warnings in a multivariate system (Dakos, 2018).

Two main approaches in the analysis of multivariate systems have been recently developed. One relies on conceiving the system as a network, where the nodes are the variables, whose dynamics are followed through time, and evaluating changes in the structure of the network. For instance, as the system moves towards a tipping point, changes in degree distributions of such a network are

representative for an approaching tipping point (Lu et al., 2021). A possible disadvantage is that the edges in the network do not have a physical foundation (Ebert-Uphoff and Deng, 2012). Recent research explores a complementary approach where causal links are calculated instead of correlation links and the strength of the causal link works as the indicator of resilience (Nowack et al., 2020).

Alternatively, dimension reduction techniques can capture overall network dynamics into a representative statistic. For instance, Principal Component Analysis (often referred to as Empirical Orthogonal Function (EOF) in climate science) can be used to get directions of change (Held and Kleinen, 2004; Weinans et al., 2019). Data can be projected onto the leading principal component, effectively yielding a univariate timeseries on which the univariate early-warning can be applied

(Held and Kleinen, 2004; Boulton and Lenton, 2015; Bathiany et al., 2013). This analysis does not make any a priori assumptions about the interactions between the different network nodes, and is therefore quite flexible in its use. However, it requires large amounts of high-quality data to yield accurate results. The underlying assumption is that as the system approaches the tipping point, the dynamics become more correlated, leading to a high explained variance of a PCA and clear

directionality in the dynamics (Lever et al., 2020).



### 4.2.4 Tipping cascades

A more peculiar challenge in the application of early-warnings is their ability to detect cascading tipping points: where a tipping point in one system has a knock-on effect on another system causing that to also tip (Klose et al., 2020; van de Leemput et al., 2018; Saade et al., 2023). Unless these systems are linked in such a way that early-warnings can be observed in both systems, the cascade is likely to present as a shock to the second system such that it would be unpredictable whilst monitoring it in isolation. For systems where tipping in one system causes the connected system partially towards a tipping point (known as a 'two-phase transition''), a stepwise jump in early-warnings in the second system can be detected. For coupled systems where the tipping in the second system happens instantaneously (a 'joint cascade') or soon after the tipping in the first system (a 'domino cascade') early-warnings are unlikely to be detectable (Klose et al., 2021).

### 4.3 Opportunities

These challenges associated with the use of early-warnings are also accompanied by a number of opportunities to improve their detection in real data. Below we outline a few of the most promising ones.

### 4.3.1 Composite metrics

Although there exists a multitude of early-warnings (CSD-based and non-CSD-based, generic and system-specific, on spatial, structural and temporal data), few studies have compared in a systematic way how these warnings behave one against the other or across different systems (Dakos et al., 2011; Veldhuis et al., 2022). Apart from the CSD-based warnings where their relationships are mathematically known (Kuehn, 2012), we simply do not know how similar information early-warnings provide. Understanding the interrelationships between all types of the most-used early-warnings will be crucial to improve their use for detecting tipping points. Composite metrics - where multiple early-warnings are combined (Drake and Griffen, 2010), abundance-based with trait-based warnings are compared (Clements and Ozgul, 2016), or machine-learning has been used to train models of multiple warnings as predictors (Brett and Rohani, 2020)- have been suggested to improve the significance and detectability of approaching tipping points. Given the increasing capacity to monitor the multivariate aspects of most systems (discussed in Sect. 4.2.3) and the increasing





availability of such data (see Sect. 4.3.2), we are not far from estimating multiple early-warnings on multiple dimensions of a system. The next step is to develop meaningful ways to best combine
them for detecting tipping points.

### 4.3.2 Increasing data availability: open databases and remote-sensed data

Over the last decade, data from long-term databases and remote-sensing has grown to become the primary sources for capturing temporal and spatial early-warnings for tipping points. Especially
for remote-sensing data, this coincides with the expansion of freely available Earth observation datasets combined with access to cloud-based systems which provide the computational power to process this increase in data (Gorelick et al., 2017). A primary focus has been on the temporal analysis of optical imagery from satellites such as the MODIS sensor (Moderate Resolution Imaging Spectroradiometer) (Liu et al., 2019; Majumder et al., 2019) or from the AVHRR sensor (Lenton et al.,
2022). Additionally, the vegetation optical depth (VOD) derived from microwave passive radiometers, has been employed to analyze early-warnings, with temporal records since the late 1970s (Smith et al., 2023; Boulton et al., 2022). Overall, the continued growth of remotely-sensed datasets is likely to drive further temporal early-warning research, while the emergence of new satellite sensors with enhanced spatial resolutions (in the order of meters) will also enable an
improved analysis of spatial early-warning at large scales. Yet, such development requires a profound understanding of the acquisition systems to effectively control and account for parameters that may impact the extraction of early-warnings.

### 4.3.3 New approaches: Machine-Learning

The success of neural networks for timeseries classification problems has inspired the development of Machine-Learning (ML) techniques for early-warnings detection. There is a natural synergy to this approach in that the same CSD phenomena manifest across a wide range of systems approaching tipping points, so the notoriously data-intensive task of training a neural network can be accomplished using plentiful synthetic data and still produce a result which can plausibly be
applied to empirical data.





Deep learning models which combine convolutional layers have been shown to outperform methods using statistical CSD-based warnings (e.g. variance, AR(1)) on a variety of both real and simulated case studies (Bury et al., 2021; Deb et al., 2022). Furthermore, these models have exhibited

success in inferring the type of upcoming bifurcation from observed pre-transition dynamics, and have performed well in extensions to phase transitions on spatiotemporal lattices (Dylewsky et al., 2023). Other ML techniques can also tell us something about how far systems are from tipping. For example, random forest models could be used to determine the factors that determine autocorrelation in forest areas on a global scale, and thus how close to tipping these forest areas

could be based on driving variables (Forzieri et al., 2022). Taken into consideration their limitations (Lapeyrolerie and Boettiger, 2022), combining ML techniques with 'traditional' early-warnings could provide some of the best prospects for monitoring systems that may be approaching tipping points.

**5. Conclusions**

The unprecedented amount of data originating from remote-sensing systems, field measurements,

surveys and simulated data, coupled with innovative models and cutting-edge computing, has made possible the development of a multitude of tools and approaches for detecting tipping points in a variety of scientific fields. Early-warnings can tell us that 'something' important may be about to happen, but they do not tell us what precisely that 'something' may be and when exactly it will happen (Dakos et al., 2015). The next step is to test the real potential of early-warnings as

preventive and management tools in anticipating natural and human-induced changes to com





**Author contributions**

VD, CB, JEB, SK developed research and drafted the paper. VD led the literature review analysed by VD, CB, JEB, JFA, DIAM, DD, CLM, BvB, LvL, EW, SK. VD, SK ran the analyses. All authors commented on the final text.

**Competing interests**

The authors declare that they have no conflict of interest

**Financial support**

CAB, JEB, and DIAM were supported by the Bezos Earth Fund via the Global Tipping Points Report project.




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
