# Peer review of "Tipping Point Detection and Early-Warnings in climate, ecological, and human systems"

_EGUsphere, 2023_

## Author Response (AR1)

**Response**

Editor's comment
The reviewers provide constructive comments that warrant a more in-depth revision, as also indicted by the authors in their response. As the manuscript under consideration is a review paper, the authors should include and briefly discuss the additional early warning methodologies mentioned by the reviewers, even if they cannot go into much detail. For example, for the methods lacking an empirical application so far, the approach such as critical speeding up could be briefly described and its merits and potential shortcomings discussed regarding potential empirical applications.

*We thank the Editor for the useful suggestions. We have now included the references to the additional metrics mentioned by reviewer 2 and requested by the editor in the discussion (in track changes) and also updated them in table 2. We have also briefly included the modelling work for cascading effects in the discussion. We hope that these changes address the points raised.*

---

## Author Response (AR2)

March 26, 2024

We thank the Editor and the reviewers for their positive assessments.
Below we respond to their comments point by point.

On behalf of all coauthors,
V Dakos
* * *
Reviewer 2
Dear authors,
Thanks for addressing most of my previous recommendations. The paper as said before is a timely contribution to the community that I believe will be useful in taking the application and uptake of EWS forward. I only have a couple of minor comments / typos listed below. That being said, it was a pleasure to read your work. It is one of those "must read" papers that I'm fortunate to review.
Best,
Juan

**Specific comments**

Line Comment
96 "There" should be "there" since the start of the sentence is: In such cases, ...
RESPONSE: Corrected

240 Perhaps it would be useful to clarify here that multimodality needs to account / control for seasonality patterns.
RESPONSE: Clarified. The text reads now:
"Multimodality in such a landscape for a given set of environmental conditions suggests that the system could exhibit alternative stable states for that range of conditions (Hirota et al., 2011; Staver et al., 2011; Abis and Brovkin, 2019; Scheffer et al., 2012b), although seasonality patterns should be accounted for to reduce misinterpretation of externally forced "states"."

510 I really like this point of the paper, but it is however misleading. For example, there has been a huge debate on whether shifts between sardines and anchovies are real regime shifts or just responses to SST which follows a periodic pattern with ENSO. In that case, the EWS on temperature is what drives the shifts in fish species, but also ENSO is dynamically independent of fish biomass. But in the Amazon, rain is not dynamically independent on tree cover. So an EWS on rain can be an indicator of the Amazon tipping because of moisture recycling feedback. Because of the feedback, the forcing (rainfall) is part of the "moving towards tipping". The notion of driver in ecology is a gray zone, it depends on the scale the system has been defined.

RESPONSE: Clarified. The text now reads:
"Yet, early-warnings of the drivers as a false-positive check make sense only in the case where the drivers are independent from the system variable. For instance, in the case of the

Amazon, early-warnings of rainfall can be seen as indicators of the Amazon tipping itself because of the strong moisture recycling feedback present, rather than an external factor inducing early-warnings on Amazon vegetation dynamics."
* * *
Reviewer 3

This manuscript provides a helpful and fairly comprehensive analysis of empirical applications of early warning signals across a variety of fields, including both biological and physical sciences. The paper provides a detailed description of the studies they chose to include and brief introductions to the methods employed by those studies. The tabulation of which methods are used in what fields, and the findings of whether those methods provided early warning or not is helpful. Ultimately, there is still a large gap between the theory of early warning indicators and there usage in real world situations, and there has been significant recent suggestions that it may be difficult to do so using using many of the initially proposed techniques (those based on summary statistics of time series). Examination of real world data on systems that underwent (and ideally did not undergo) critical transitions is therefore a key first step to understanding the performance of different types of indicators, and this paper represents the first such effort to my knowledge. The database of studies assembled here is an excellent resource that could even be expanded, and perhaps one day this could even lead to a database of time series exhibited critical transitions that could be analyzed in a comprehensive way. This paper is well done and I do not see any central conceptual issues that would lead me to object to publishing it.

However, I have one suggestion for a potential improvement to the manuscript which could improve the paper if implementable. There is a decent discussion on how biases could impact the practical application of early warning signals- and I think these are extremely critical issues for the practical use of early warning indicators. However, I'm still concerned that bias could have an impact on how the data presented in this paper is interpreted. In particular, each paper in the study uses different statistical methods, and methods based on hypothesis testing and significance thresholds could be influenced by p-hacking. It's possible that papers that didn't find early warning indicators before critical transitions were on average more careful in their data analyses. I checked several papers in the database (at random from papers finding early warning and comprehensively from those that didn't, as the latter were fewer in number). While I lack the time to do this closely, I did notice that several of the papers with negative findings used a statistical method (Bayesian changepoint analysis) that wasn't used in any of the papers with positive findings that I checked. I would be curious if the authors have thought about (or somehow quantified) the variance in statistical tools between studies and if they are concerned this variance could introduce a systematic error or they think it doesn't matter. Either way, collecting such a database is a requirement for performing further quantitative studies of the appearance of early warning indicators across a variety of systems, and I think it is very helpful and important for this paper and database to be available, therefore I recommend publication.

RESPONSE: We have now expanded on this point in the discussion.

"One important aspect that we have not considered in the comparative analysis of the reviewed literature is the fact that each paper uses different statistical methods, different hypothesis testing approaches (like surrogate data, Bayesian and frequentist p-values) and different significance levels to conclude on the identification of an early-warning or not. To what extend such differences may even induce p-hacking is unclear, but needs to be acknowledged in future work."

Two small editorial suggestions:

Line 57 Page 2: "did not allow to anticipate" to "did not allow anticipation of"
RESPONSE: Corrected

Line 79-80 Page 3: "Yet, although the utility of early warnings has led to early-warnings proliferating beyond ecology and climate and have been
applied across a variety of scientific domains," I would change to "Yet, although the utility of early warnings has led to early-warnings proliferating beyond ecology and climate and beeing applied across a variety of scientific domains"
RESPONSE: Corrected